# ASYNCHRONOUS GRAPH GENERATORS

## ABSTRACT

We introduce the asynchronous graph generator (AGG), a novel graph neural network architecture for multi-channel time series which models observations as nodes on a dynamic graph and can thus perform data imputation by transductive node generation. Completely free from recurrent components or assumptions about temporal regularity, AGG represents measurements, timestamps and metadata directly in the nodes via learnable embeddings, to then leverage attention to learn expressive relationships across the variables of interest. This way, the proposed architecture implicitly learns a causal graph representation of sensor measurements which can be conditioned on unseen timestamps and metadata to predict new measurements by an expansion of the learnt graph. The proposed AGG is compared both conceptually and empirically to previous work, and the impact of data augmentation on the performance of AGG is also briefly discussed. Our experiments reveal that AGG achieved state-of-the-art results in time series data imputation, classification and prediction for the benchmark datasets *Beijing Air Quality*, *PhysioNet Challenge 2012* and *UCI localisation*.

## 1 INTRODUCTION

Incomplete time series data are ubiquitous in a number of applications (Miao et al., 2019), including medical logs, meteorology records, traffic monitoring, financial transactions and IoT sensing. Missing records may be due to various reasons which include failures either in the acquisition or transmission systems, privacy protocols, or simply because the data are collected asynchronously in time. Missing data is an issue in itself but also hinders applications, for example, the public dataset PhysioNet (Silva et al., 2012) has a 78% average missing rate which makes it challenging to extract useful information from the dataset for, e.g., for predicting mortality. In this setting, imputation refers to filling in the missing values using the available sparse observations (Little & Rubin, 2019), and can be achieved by methods that exploit both temporal and spatial dependencies (Yoon et al., 2017; Yi et al., 2016).

Existing approaches (Cao et al., 2018) to imputation in multi-sensor time series often assume temporal regularity of the data, which is a consequence of representing the values of the series through a matrix with missing entries as shown in Fig. 1a. This representation implicitly produces two critical assumptions: i) the notion of order (causality), e.g., $x_1$ precedes $x_2$, and ii) a fixed sampling rate implying synchronous data acquisition. We assert that this representation is detrimental to successfully learn latent dynamics generating the (sparse) observations, therefore, we propose to relax these stringent assumptions and represent observations as nodes in an asynchronous directed graph, such as that depicted in Fig. 1b. This approach is robust to the occurrence of missing data and exploits the permutation invariance of multiple sensors to perform imputation as a transductive node generation operation over graph embeddings as depicted in Fig. 1c. We refer to the proposed representation as asynchronous graph generator (AGG).

Deep-learning-based approaches to imputation of missing data have become increasingly popular in the last five years (Yoon et al., 2018b;a; Liu et al., 2019; Cao et al., 2018). However, in general these methods rely on slight modifications of standard neural architectures tailored for discrete-time complete data and are thus unable to fully incorporate available relational information related to, e.g., temporal, spatial or operating conditions (Bai et al., 2018; Chung et al., 2014). We argue that continuous-time graphs are a promising resource for incorporating stronger inductive biases in the analysis of multivariate signals, in particular with applications to data imputation. We assume no data regularity beyond what is explicitly observed through each sensor, all with the aim to learn the

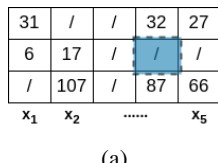 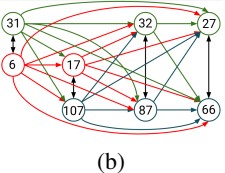 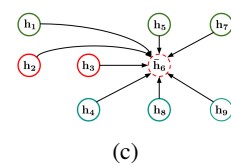

|     (a)     |     (b)     |     (c)     |

Figure 1: (a) Matrix time-series representation (Cao et al., 2018). (b) Asynchronous directed graph representing observations and causal relationships through directed edges; colours represent different metadata encoding. (c) Imputation performed by generating a new nodes, in this case, node $\bar{h}_6$

latent dynamics as agnostically as possible. Using an asynchronous graph is pivotal to fulfil this aim as it allows us to identify expressive relationships among measurements in large and incomplete sensor networks, as those found in real-world applications.

## 2 RELATED WORK

The literature addressing missing value imputation in time series is vast. Enormous work has been dedicated to attempting imputation using classical (non-deep learning) approaches (Beretta & Santaniello, 2016; Troyanskaya et al., 2001; Ghahramani & Jordan, 1993; Nelwamondo et al., 2007; Durbin & Koopman, 2012; Kihoro et al., 2013; Cichocki & Phan, 2009; Cai et al., 2011; Rao et al., 2015; Mei et al., 2017; Yu et al., 2016; Yi et al., 2016).

More recently, deep learning models have been successfully developed for multi-sensor time series imputation, in particular, using recurrent neural networks (RNNs) (Cao et al., 2018; Yoon et al., 2018b; Lipton et al., 2016; Che et al., 2018; Luo et al., 2018). Notably, GRU-D (Che et al., 2018) analyses sequences with missing data by controlling the decay of the hidden states of a gated RNN, while BRITS (Cao et al., 2018) implements a bidirectional GRU-D that incorporates cross-channel correlation to perform spatial imputation. These RNN-based methods assume temporal regularity of data, i.e., a fixed sampling rate.

Adversarial strategies have also been applied to imputation. GAIN (Yoon et al., 2018a) uses GANs (Goodfellow et al., 2020) to perform imputation in the i.i.d. setting where dependencies among sensors are neglected, while Luo et al. (2018; 2019) trains models to generate realistic synthetic sequences. Miao et al. (2021) used an approach similar to GAIN but conditioned the generator on the predicted label to reconstruct missing values. Lastly, Liu et al. (2019) addressed the imputation problem for multi-scale highly-sparse series using hierarchical models.

Concurrently, graph neural networks (GNN) have found applications in spatio-temporal forecasting, where the idea underpinning most methods is the extension of RNN architectures to the graph domain. For instance, Seo et al. (2018) implemented GRU cells as nodes combined with spectral GNN operations (Defferrard et al., 2016), while Li et al. (2018) replaced spectral GNNs with a diffusion-convolutional network (Atwood & Towsley, 2016). Scarselli et al. (2008); Li et al. (2016); Yu et al. (2017); Wu et al. (2019; 2020) propose, instead, spatio-temporal graph convolutional networks that alternate convolutions on temporal and spatial dimensions. Similar approaches have focused on spatio-temporal data by combining Transformer-like architectures with RNNs (Cai et al., 2020; Zhang et al., 2018). Temporal graph networks (Rossi et al., 2020; Cini et al., 2022) learn node embeddings in dynamical graphs but again heavily relying on RNNs to extract temporal encodings. Lastly, recent works used GNNs for imputation of missing features in the i.i.d. case: Spinelli et al. (2020) trained GNNs for the data reconstruction task, while You et al. (2020) proposed a bipartite graph representation for feature imputation.

To the best of our knowledge, no previous GNN-based method approaches the imputation problem from the perspective of an asynchronous graph. They rely on RNNs in some form and thus implicitly adopt the strong assumptions about sample regularity as a consequence.

## 3 THE AGG ARCHITECTURE

Asynchronous graphs are a subclass of continuous-time dynamic graphs (CTDG) and are generally represented as a timed list of events, i.e., operations over edges and nodes including addition, deletion or feature transformations (Rossi et al., 2020). The proposed AGG considers each new sensor measurement as an expansion of the graph—or node additions—with the directed edges representing the temporal (causal) relationship among new and past measurements. Being a sequence of time-stamped events, we denote the graph by $\mathcal{G} = \{x_1, x_2, \ldots\}$.

The main objective of AGG is to perform transductive node generation, that is, given a set of observations composed of values, timestamps and additional measurements referred to as *metadata*, AGG generates the value for a set of new nodes conditional on any timestamp and metadata. We emphasise the timestamps need not be uniformly sampled or even ordered.

Transductive node generation, as seen in Fig. 1c, is a node addition to the existing asynchronous graph. When a node is added to a graph—which is permutation invariant (Bronstein et al., 2021)—it has no notion of position but only relationship to other nodes via edges. It is through the temporal encoding that we condition the node to have the idea of order within the graph. If the encoding places the new node within the temporal "neighbourhood" of the other nodes in the graph, we refer to data imputation, whereas if the new node comes after the known temporal encodings we refer to prediction. Furthermore we can condition the graph to generate nodes with continuous values (regression) or discrete values (classification). We can see that the class of node generation is arbitrary and, given a flexible notion of encoding, allows the AGG to used for a wide variety of tasks from imputation to anomaly detection.

Data imputation can also be seen as a type of self-supervised pre-training through masked data augmentation (Balestriero et al., 2022). After performing imputation, the graph embeddings can leverage their expressive representation for regression, classification and even anomaly detection in the same way that masked pre-training is leveraged in architectures like BERT (Devlin et al., 2019). Our self-supervised approach splits observations into inputs and targets—see Fig. 2—to then organise them into batches for training a graph attention-based architecture. We next present the data treatment and the proposed architecture.

### 3.1 PROBLEM FORMULATION AND DATA PREPARATION

For clarity of presentation, we assume the existence of continuous-time latent signals which are only measured through a finite set of observations $\mathcal{D} = \{x_n\}_{n=1}^{N}$. The $i$-th measurement is given by

$$x_n = [y_n, t_n, m_n] \in \mathbb{R}^{d_y + 1 + d_m}, \tag{1}$$

where $y_n \in \mathbb{R}^{d_y}$ is the **value**, $t_n \in \mathbb{R}$ is the **timestamp** and $m_n \in \mathbb{R}^{d_m}$ is all the available **metadata** including—but not limited to—type, location and operating conditions of the measurement. Our aim is to extract knowledge from $\mathcal{D}$ to predict **values** corresponding to a set of **timestamps** and **metadata** previously unseen. To exemplify the role of this notation consider the Beijing dataset, where **metadata** captures the measurements' type (e.g., PM2.5, pressure, temperature) as well as their location. Our formulation stems from the assumption that values across the graph are related not only by their timestamps but also by additional features such as channel id and sensor location. Explicitly encoding this metadata in the nodes allows the graph to learn in a way that exploits the interactions among the relevant variables, e.g., sensors of different types should interact differently as should different physical locations. Our hypothesis is that by encoding this metadata the graph can be fully context aware and thus performant.

The process of leveraging the data to train AGG is described next, refer to Fig. 2 for an illustration of a particular case. First, the dataset $\mathcal{D}$ in equation 1 is obtained via an acquisition system (Fig. 2a) and each measurement is considered as a node in a graph. Then, we order the nodes wrt their timestamps and randomly split the dataset into input and target samples (blue and red in Fig. 2b). Lastly, the dataset is divided into samples of $L$ inputs and 1 output by sequentially passing through the observations with a stride of $\Delta$ (Fig. 2c).

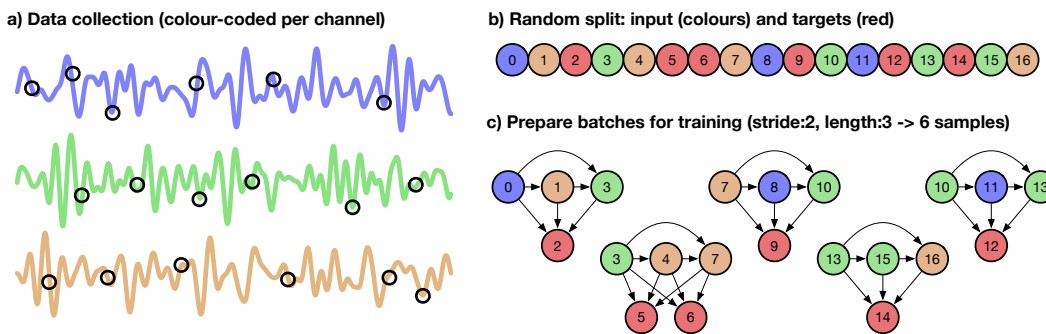

Figure 2: Illustration of the data preparation process to train AGG for a 3-channel signals (colour-coded) with $n = 17$ observations, $\approx 35\%$ of samples removed (red), block length $L = 3$ and stride $\Delta = 2$. There are 6 samples in this batch, where the targets 5 and 6 constitute 2 separate samples.

## 3.2 LEARNABLE EMBEDDINGS FOR VALUE, TIME-STAMPS AND METADATA

**Temporal embedding.** Graphs are naturally permutation invariant so in order to learn flexible representations of temporal differences, such as periodicities and long-range dynamics, we must encode the temporal position along with nodes features. Following Kazemi et al. (2019), we use the learnable temporal encoding **t2v** and then use these learnt representation in a similar vein as positional encoding in Vaswani et al. (2017). For a $x_n$ as defined in equation 1, this embedding is parametrised as

$$\mathbf{t2v}(\tau_n) = [\omega_0\tau + \varphi_0, \mathcal{F}(\omega_1\tau_n + \varphi_1), \mathcal{F}(\omega_2\tau_n + \varphi_2), \ldots, \mathcal{F}(\omega_{D_t-1}\tau_n + \varphi_{D_t-1})]^\top \in \mathbb{R}^{D_t}, \tag{2}$$

where $\tau_n$ is the temporal difference between $x_n$ and last-observed node $x_N$, i.e., $\tau_n = t_N - t_n \geq 0$; $\{\omega_k\}_k$ and $\{\varphi_k\}_k$ are learnable parameters; and $\mathcal{F}$ is a periodic function. Inspired by Kazemi et al. (2019), we choose $\mathcal{F}(\cdot) = \sin(\cdot)$ in all implementations of AGG.

**Metadata embedding.** In order to utilise measurements of different nature (defined by the metadata) one could be tempted to represent all interactions via a heterogeneous graph and build specific models for each interaction of nodes and edges. However, this would require us to cater for all possible relationships among nodes with minimal weight sharing throughout the model. To circumvent this challenge, AGG is modelled as a homogeneous graph instead, where a single learnable form of interaction operates over values $y_n$, time stamps $t_n$ and metadata $m_n$ provided by the sensor measurement. In the same vein as the temporal embedding, the metadata is represented by a set of learnable embeddings, a practice that has become prevalent in the field of natural language programming for *learnable word embeddings* beginning with Bengio et al. (2000). This way, we aim to include all available information as a form of inductive bias (Bronstein et al., 2021) into the model, and leave the graph structure to exploit rich relationships among features and values via an attention mechanism.

AGG builds metadata embeddings based on whether they are discrete or continuous: discrete metadata (e.g. categorical data) are embedded via hashing, that is, a matrix of learnable weights is sliced at the index of the relevant category. Similarly, continuous metadata is embedded into higher dimensions through a learnable projection matrix. The complete embedding of the metadata (considering both discrete and continuous parts) is denoted $\text{embed}(m_n) \in \mathbb{R}^{D_m}$

To enhance the representation power of the overall architecture, we follow Veličković et al. (2018) and also include a learnable projection for the value denoted $\text{embed}(y_n) \in \mathbb{R}^{D_y}$. Thus the AGG is a heterogeneous graph $\mathcal{G}$ with $n$-th node containing

$$h_0 = \text{Concat}\left[\text{embed}(y_n), \mathbf{t2v}(\tau_n), \text{embed}(m_n)\right] \in \mathbb{R}^{D_y+D_t+D_m}, \tag{3}$$

where the explicit dependence on the index $n$ is dropped unless necessary.

Observe that we denoted the original dimensions in lowercase ($d_y$ and $d_m$) and the embedded ones in uppercase ($D_y$, $D_t$ and $D_m$). Also, following equation 3 we define $d_{\text{encoder}} = \dim(h_0) = D_y + D_t + D_m$, where the notation $h_0$ will be clarified in the next section.

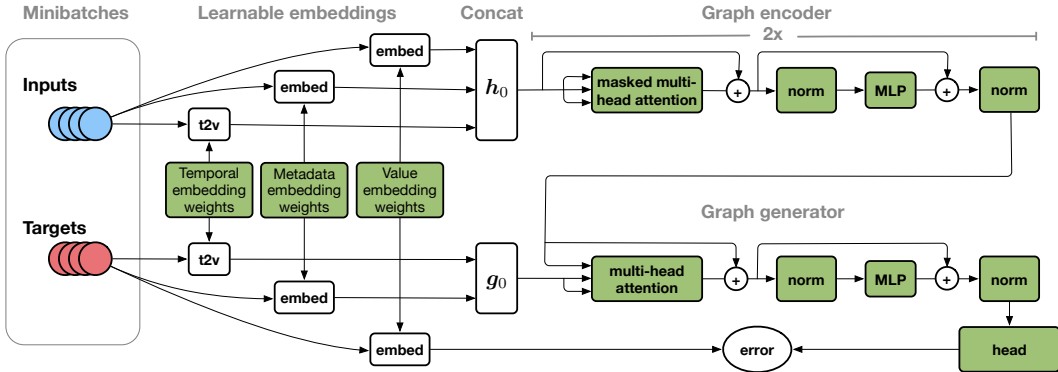

Figure 3: AGG architecture: The sections of the network are indicated at the top of the figure. Inputs and target are represented as blue and red circles respectively, fixed operations are denoted by white blocks and learnable transformations in green blocks.

Fig. 3 illustrates the embedding procedure under the title **learnable embeddings**. The embeddings then enter a sequence of encoder and decoder blocks comprising attention and fully connected layers with layer-norms and skip connections through addition. The next two sections present the encoder and the generator stages.

### 3.3 ASYNCHRONOUS GRAPH ENCODING

Towards improved performance and expressibility (Brody et al., 2022; Veličković et al., 2018; Vaswani et al., 2017), the encoder features a multi-head self-attention layer representing the interactions among values, timestamps, and metadata.

Following equation 3, for a single node we denote $h_{i-1}$ and $h_i$ the input and output of the $i$-th encoder block respectively ($i \geq 1$). However, recall from Sec. 3.1 that AGG takes $L$ nodes simultaneously, thus, we denote $\boldsymbol{h}_i$ as the concatenation of the $h_i$'s coming from these $L$ nodes. Therefore, each $\boldsymbol{h}_i \in \mathbb{R}^{L \times d_{\text{encode}}}$ is a tensor comprising $L$ node embeddings.

The $j$-th head of the $i$-th attention layer is thus given by:

$$\text{Attention}(\boldsymbol{Q}_{ij}, \boldsymbol{K}_{ij}, \boldsymbol{V}_{ij}) = \text{softmax}(\boldsymbol{M} \circ \boldsymbol{E}_{ij})\boldsymbol{V}_{ij} \in \mathbb{R}^{L \times d_v}, \tag{4}$$

where $\circ$ is the Hadamard (or element-wise) product and

- $\boldsymbol{Q}_{ij} = \boldsymbol{h}_{i-1}\boldsymbol{W}_j^Q \in \mathbb{R}^{L \times d_q}$, $\boldsymbol{K}_{ij} = \boldsymbol{h}_{i-1}\boldsymbol{W}_j^K \in \mathbb{R}^{L \times d_k}$, $\boldsymbol{V}_{ij} = \boldsymbol{h}_{i-1}\boldsymbol{W}_j^V \in \mathbb{R}^{L \times d_v}$ are the query, key and value embeddings respectively.

- $\boldsymbol{W}_i^Q \in \mathbb{R}^{d_{\text{encoder}} \times d_q}$, $\boldsymbol{W}_i^K \in \mathbb{R}^{d_{\text{encoder}} \times d_k}$, $\boldsymbol{W}_i^V \in \mathbb{R}^{d_{\text{encoder}} \times d_v}$ are the projection matrices.

- $[\boldsymbol{M}]_{qk} = \mathbf{1}_{t_q \leq t_k}$ is a temporal mask ensuring the operation of AGG is over *causal* graphs. Dropout (Hinton et al., 2012) is applied to the mask during training to promote sparsity and redundancy in the graphs representation by randomly severing connections.

- $\boldsymbol{E}_{ij} = d_k^{1/2}\boldsymbol{Q}_{ij}\boldsymbol{K}_{ij}^{\top} \in \mathbb{R}^{L \times L}$ is the dot product attention Vaswani et al. (2017) matrix which is equivalent to a fully connected weighted graph (Veličković, 2023) pruned via $\boldsymbol{M}$. Under the graph interpretation, $\boldsymbol{E}$ is the weighted adjacency matrix for the $L$ nodes in the asynchronous graph, where the weight represents the relevance of neighbouring nodes in determining the features of any other node.

Then, the $i$-th multihead attention layer is simply the weighted concatenation of its attention heads:

$$\text{MultiHead}_i = \text{Concat}\left[\text{Attention}(\boldsymbol{Q}_{i1}, \boldsymbol{K}_{i1}, \boldsymbol{V}_{i1}), \ldots, \text{Attention}(\boldsymbol{Q}_{il}, \boldsymbol{K}_{il}, \boldsymbol{V}_{il})\right]\boldsymbol{W}^O \in \mathbb{R}^{L \times d_{\text{encode}}}. \tag{5}$$

Lastly, the output of the $i$-th multi-head attention is normalised via a layer normalisation (Ba et al., 2016) followed by a multi-layer perceptron (MLP). The MLP consists of a 2-layer feed forward

network with a LeakyReLU (Maas et al., 2013) activation and Dropout (Hinton et al., 2012) in the hidden layer, followed by a linear activation layer. The MLP has layer sizes of $[d_{\text{encode}}, l \times d_{\text{encode}}, d_{\text{encode}}]$, with $l$ is the number of heads. Throughout each block there is extensive use of skip connections following inspiration from the Transformer (Vaswani et al., 2017) and the original introduction of the residual connections, ResNet (He et al., 2016).

The output of the $i$-th block is then calculated by:

$$\boldsymbol{u}_i = \boldsymbol{h}_{i-1} + \text{MultiHead}_i \tag{6}$$

$$\boldsymbol{h}_i = \text{LayerNorm}\left[\boldsymbol{u}_i + \text{MLP}\left(\text{LayerNorm}\left[\boldsymbol{u}_i\right]\right)\right]. \tag{7}$$

Therefore, equations 4 - 7 completely define the sequence of outputs from the asynchronous graph encoder blocks $\boldsymbol{h}_0, \dots, \boldsymbol{h}_l$.

### 3.4 ASYNCHRONOUS GRAPH GENERATION

AGG leverages cross attention—see Fig. 3—between the output of the last asynchronous encoder block $\boldsymbol{h}_l$ and the concatenation of (target) temporal/metadata embeddings for conditional generation, the latter denoted by

$$\boldsymbol{g}_0 = \text{Concat}[\textbf{t2v}(\tau_t), \text{embed}(m_t))] \in \mathbb{R}^{d_g}, \tag{8}$$

where $d_g = D_m + D_t$. Transductive node generation, conditioned on the timestamps and metadata, defines where in the graph the new node should be located.

Conditional generation also leverages multiple attention heads, which, akin to equations 4 & 5, is given by

$$\text{CrossMultiHead} = \text{Concat}\left[\text{Attention}(\boldsymbol{Q}_1, \boldsymbol{K}_1, \boldsymbol{V}_1), \dots, \text{Attention}(\boldsymbol{Q}_l, \boldsymbol{K}_l, \boldsymbol{V}_l)\right] \boldsymbol{W}^O \in \mathbb{R}^{L \times d_{\text{encode}}}, \tag{9}$$

where

- $\boldsymbol{Q}_j = \bar{\boldsymbol{g}}_0 \overline{\boldsymbol{W}}_j^Q \in \mathbb{R}^{d_{\bar{q}}}, \boldsymbol{K}_j = \bar{\boldsymbol{h}}_l \overline{\boldsymbol{W}}_j^K \in \mathbb{R}^{L \times d_k}, \boldsymbol{V}_j = \bar{\boldsymbol{h}}_l \overline{\boldsymbol{W}}_j^V \in \mathbb{R}^{L \times d_v}$ are the query, key and value respectively, and $\bar{\boldsymbol{g}}_0 = \text{LayerNorm}[\boldsymbol{g}_0]$ and $\bar{\boldsymbol{h}}_l = \text{LayerNorm}[\boldsymbol{h}_l]$.
- $\overline{\boldsymbol{W}}_j^Q \in \mathbb{R}^{d_g \times d_{\bar{q}}}, \overline{\boldsymbol{W}}_j^K \in \mathbb{R}^{d_{\text{encode}} \times d_k}, \overline{\boldsymbol{W}}_j^V \in \mathbb{R}^{d_{\text{encode}} \times d_v}, \boldsymbol{W}^O \in \mathbb{R}^{L \times d_g}$ are the projection matrices.
- $\boldsymbol{E}_j = d_k^{1/2} \boldsymbol{Q}_j \boldsymbol{K}_j^T \in \mathbb{R}^L$.

**Remark.** *The cross attention block does not include a causal mask, it implements a fully connected attention graph over all embeddings; $\boldsymbol{M} = \boldsymbol{1}$; Dropout is applied during training.*

Additionally, similar to the asynchronous encoder block, the generator follows the cross attention layer with a set of LayerNorms, skip connections and an MLP, such that:

$$\bar{\boldsymbol{u}} = \boldsymbol{g}_0 + \text{CrossMultiHead} \tag{10}$$

$$\boldsymbol{g}_1 = \text{LayerNorm}\left[\bar{\boldsymbol{u}} + \text{MLP}\left(\text{LayerNorm}\left[\bar{\boldsymbol{u}}\right]\right)\right]. \tag{11}$$

Lastly, depending on the task, we use the generated decoding $\boldsymbol{g}_1$ and fit a trainable head to purpose, e.g., a classification head or a regression head, which consists of an MLP that projects $\boldsymbol{g}_1$ to the desired value $\hat{y}_n$, such that:

$$\hat{y}_n = \text{MLP}(\boldsymbol{g}_1). \tag{12}$$

**Remark.** *Preliminary experimental evaluation of AGG using a single generator block as presented here provided satisfactory results. The choice to maintain this architecture follows Occam's razor.*

Fig. 3 shows a diagram of the entire AGG architecture identifying the connections, inputs, targets, as well as fixed and trainable blocks.

## 4 RELATIONSHIP TO PREVIOUS METHODS

Our work is conceptually closer to those of Cini et al. (2022); Rossi et al. (2020) albeit with some key differences. They propose bidirectional RNNs encapsulated in GNNs, where a series of RNNs

are interconnected through gates controlled by message passing NNs. These works consider the time series as a sequence of weighted directed graphs, thus assuming each node to be identified and labelled with a unique id and consistently available at all evenly-sampled timestamps. Therefore, their graphs have a fixed topology over time and thus the methods operate mainly by exploiting of network homophily. Furthermore, the temporal dynamics are firmly delegated to the RNN, as a consequence, the known drawbacks of RNNs hinder the applicability of the methods for imputation, namely long-term memory retention and temporal dependencies, vanishing gradients, memory staleness, hidden-state bottleneck, to name a few (Rossi et al., 2020).

The proposed AGG does not use recurrent architectures and learns long-term dependencies directly via a graph over the nodes features (measurements). The node features are embedded into a high-dimensional space to represent their position in space and time, then their relationships are captured by a learnable graph whose connections are defined via conditional dot product attention. Additionally, the causal relationship of the nodes is enforced through the masked attention mechanism. The AGG has no recurrence so memory staleness (Rossi et al., 2020) is inherently avoided and, as a consequence, the range of temporal dependencies that can be learnt are only limited by the context window of the AGG input sequence and not the model. A critical feature of AGG that should not be overlooked is its ability to leverage past measurements of adjacent sensors, which we believe to be a significant shortcoming of recurrent message-passing neural networks proposed by Rossi et al. (2020), then expanded by Cini et al. (2022). The AGG, on the other hand, is able to look at past measurements of adjacent sensors in order to compute each node embedding, this is a key component to encoding both the *coherence* and *phase* relationship (Granger, 1969), which quantify the similarity and delay between a pair of time series. We argue that models that consider time series as a set of sequential graphs ignore the coherence and phase components of a dynamic system, while by leveraging attention over past measurements of adjacent nodes the AGG is able to effectively capture the phase and coherence dynamics of the system as a whole.

## 5 EXPERIMENTAL EVALUATION

**Benchmark models and datasets.** AGG was compared against state-of-the-art models SSGAN (Miao et al., 2021), BRITS (Cao et al., 2018), NAOMI (Liu et al., 2019), GP-VAE (Fortuin et al., 2020) on three datasets for imputation: the *Beijing Air Quality* (Yi et al., 2016), *PhysioNet Challenge 2012* (Silva et al., 2012) and *UCI Localization Data for Person Activity* (Kaluža et al., 2014). The first two datasets were also used for classification and regression of mortality and PM2.5 respectively. All data were standardised per channel. See Appendix A.1 for additional details.

**Implementation details.** A common AGG architecture was implemented without hyper-parameter tuning for all datasets. We considered two encoder layers (Sec. 3.3) and one generator layer (Sec. 3.4), followed by a regression or classification head depending on the task. All embeddings were 16 dimensions per feature with 8 attention heads. The MLPs in equations 7 and 11 featured 2 layers: an input layer of dimension $5 \times 16 = 80$ and a hidden layer of dimension equal to number of heads $\times$ embedding dimension $= 8 \times 80 = 640$, which was then reduced back to the embedding dimension (80). During training, we used a Dropout rate of 0.2 for both the MLP layers and the attention masking. As a result, the model has 378k trainable parameters with a standard context length of $L = 100$ nodes, which are padded if the context length exceeds the dataset such as some samples in the *Physionet* dataset. Refer to Fig. 3 for more details of the AGG architecture.

**Infrastructure.** AGG was implemented on PyTorch (Paszke et al., 2019) using an Nvidia RTX Titan GPU with 24GB of VRAM and 4608 CUDA Cores, and an Intel Core i9-9900K with 16 cores and 32GB of RAM running Ubuntu 22.04 64bit. Code is available[1].

### 5.1 DATA IMPUTATION

Following Miao et al. (2021), we addressed the unsupervised imputation task by randomly splitting the data into $r\%$ for targets and $(1 - r)\%$ for inputs (see Figs. 2 and 3), with the targets split again in $80\% - 20\%$ for training and validation respectively. We chose $r \in \{10, 30, 50, 70, 90\}$ and evaluated the imputation performance using the Root Mean Square Error (RMSE). This setting replicates an

---

[1]https://github.com/▨▨▨▨▨▨▨▨

Table 1: Time series imputation performance (RMSE) for all models considered under different percentage of removed data ($r$). Improvement denotes (as a percentage): AGG vs SSGAN.

| Dataset | Removed ($r$) | Mean | GP-VAE | NAOMI | BRITS | SSGAN | AGG | Improvement |
|---|---|---|---|---|---|---|---|---|
| | 10% | 0.813 | 0.670 | 0.641 | 0.621 | 0.600 | **0.195** | 67.5% |
| | 30% | 0.873 | 0.726 | 0.724 | 0.686 | 0.666 | **0.221** | 66.8% |
| UCI | 50% | 0.933 | 0.796 | 0.794 | 0.786 | 0.759 | **0.222** | 70.8% |
| | 70% | 0.943 | 0.846 | 0.854 | 0.836 | 0.803 | **0.234** | 70.9% |
| | 90% | 0.963 | 0.882 | 0.897 | 0.867 | 0.841 | **0.241** | 71.3% |
| | 10% | 0.799 | 0.677 | 0.632 | 0.611 | 0.598 | **0.494** | 17.4% |
| | 30% | 0.863 | 0.707 | 0.703 | 0.672 | 0.670 | **0.535** | 20.1% |
| PhysioNet | 50% | 0.916 | 0.787 | 0.783 | 0.779 | 0.762 | **0.532** | 30.2% |
| | 70% | 0.936 | 0.837 | 0.835 | 0.809 | 0.782 | **0.589** | 24.7% |
| | 90% | 0.952 | 0.879 | 0.865 | 0.850 | 0.818 | **0.702** | 14.2% |
| | 10% | 0.763 | 0.522 | 0.522 | 0.531 | 0.435 | **0.176** | 59.5% |
| | 30% | 0.806 | 0.562 | 0.558 | 0.561 | 0.461 | **0.157** | 65.9% |
| Beijing | 50% | 0.866 | 0.602 | 0.602 | 0.581 | 0.490 | **0.197** | 59.8% |
| | 70% | 0.898 | 0.709 | 0.701 | 0.641 | 0.603 | **0.225** | 62.7% |
| | 90% | 0.919 | 0.771 | 0.762 | 0.720 | 0.660 | **0.329** | 50.2% |

Table 2: Performance of pre-trained models on classification (left) & regression (right)

| Method | PhysioNet ICU mortality (AUC) | Beijing PM2.5 regression (MAE) |
|---|---|---|
| GRIN | N/A | 10.23 |
| BRITS | $0.850 \pm 0.002$ | 11.56 |
| **AGG** | $\mathbf{0.862 \pm 0.0075}$ | **3.64** |

extremely-sparse imputation scheme, to be addressed via transductive node generation (Fig. 1c). See Appendices A.1.1 for details about the Beijing dataset and A.2 for data removal and batching.

Table 1 shows the performance of the methods considered, alongside the baseline Mean imputation method and AGG's performance improvement over current state-of-the-art SSGAN. Across all values of removed data ($r$), AGG outperformed all benchmarks and exhibited an average improvement of 21.3% on PhysioNet, 59.6% on Beijing PM2.5 dataset, and 69.5% on UCI (wrt SSGAN). A keen observer would note that unlike past methodologies the AGG does not decrease its performance monotonically with $r$, in fact under some circumstances it improves with $r$ (note the improvement of $r = 30\%$ vs $r = 10\%$ on the Beijing dataset). We attribute this behaviour to two key characteristics of the AGG, the first being the invariance of the architecture to sparsity of the data, such that the model sees little change in the underlying signal with $r \leq 50\%$. The second is the sensitivity of the AGG to data augmentation (see Sec. 6): it seems that $r = 30\%$ is an inflection point whereby there has been sufficient data removed to properly train AGG but not enough that the information (in an *information theoretic* (Shannon, 1949) sense) of the underlying dynamics has been diminished.

## 5.2 CLASSIFICATION AND REGRESSION

Following the methodologies of Cao et al. (2018); Miao et al. (2021), the model pretrained on the imputation task was used to predict in-hospital mortality on Physionet. Specifically, we fine-tuned the model pretrained with $10\%$ of data removed as explained above and, similarly to BRITS, we performed $k$-fold ($k = 5$) cross validation with the entire dataset. AGG achieved an average **AUC = 0.862**, thus improving over BRITS which reported AUC = 0.850 (Silva et al., 2012). Though SSGAN did not report an exact performance index for this experiment, from Fig.4a in Miao et al. (2021) SSGAN appeared to perform on par with BRITS with AUC $\simeq 0.85$.

AGG was then used to predict PM2.5 (Beijing dataset) and compared against the two best-scoring methodologies encountered in the literature following the setting in Yi et al. (2016) regarding the test/train split and the use of MAE. AGG scored a PM2.5 prediction **MAE = 3.64** thus outperforming both BRITS (Cao et al., 2018) and GRIN (Cini et al., 2022) as showed in Table 2. We conjecture that the considerable improvement of AGG (64.4%) wrt GRIN can be explained by its strong in-

ductive bias resulting from the spatial encoding, which captures the inner dynamics of spatially and temporally correlated data, thus effectively learning the phase shift among locations.

## 6 DISCUSSION: ON THE EFFECTIVENESS OF DATA AUGMENTATION

Conceptually, the distinguishing features of the AGG are its invariance to sparsity (missing data) and its ability to exploit translation equivariance of the signal. It is widely accepted that data augmentation regularises a model towards the transformations that are applied (Balestriero et al., 2022; Neyshabur, 2017; Neyshabur et al., 2014). If these transforms align with the geometric priors (Bronstein et al., 2021) they can be exploited to can create a much more expressive representation of features in the signal space. This would allow the model to capture relevant interacting dynamics between channels, while ignoring superfluous information. It

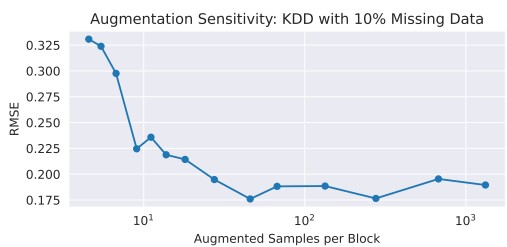

Figure 4: AGG performance (RMSE) vs number of training samples produced from the *same* dataset through augmentation.

is expected that this inductive bias introduces some form of capacity control (Neyshabur, 2017) which in turn allows for successful generalisation.

Data augmentation should then emphasise geometric priors in our model to fully learn a generalisable representation of the signal of interest. Our choice of augmentation is inspired by self supervised learning (SSL) (Misra & Maaten, 2020; Zbontar et al., 2021) in computer vision, where augmentations exploit the translation equivariance in images through shift operations. In the same vein, we randomly remove samples from the training set to promote sparsity in our dataset and shift the inputs (relative to targets) in order to leverage the translation equivariance.

We studied the effect of this approach to data augmentation on the imputation task with $10\%$ of the data removed (as defined in Sec. 5). To this end, we varied the stride length of each sample: the finer the stride, the more data samples are generated from the same training data—more details in Appendices A.2 and A.3. Fig. 4 shows the effect of the number of augmented samples of each block on the imputation performance via RMSE over the validation set, as defined by Yi et al. (2016).

The validation RMSE of AGG decreased sharply up to approximately 60x augmented samples, thus confirming the existence of a threshold for data augmentation in AGG after which complexity cost increases without gain in performance. This is consistent with Balestriero et al. (2022) who found empirically that 50x augmented samples were required to estimate their closed form of the loss. In general cases this threshold should be determined based on the sampling theorem (Shannon, 1949), which relates the observation rate with the dynamic content of the signals (for the stationary case).

## 7 CONCLUSIONS

We have presented asynchronous graph generators (AGGs), a family of attention-based models for multichannel time series that represents observations as nodes of a dynamic graph without assuming temporal regularity or recurrence. Using data-augmentation techniques inspired from computer vision and learnable embeddings from language models, we have shown that AGG can be successfully trained under missing-data regimes to discover rich relationships among variables of interest. Once trained, AGG can be used for data imputation—and as a consequence classification and prediction—by means of a conditional transductive node generation operation, that is, by generating a new node in the graph at a given timestamp (and metadata). We have experimentally validated the superiority of AGG against the state of the art on three relevant datasets and different rates of missing values. Our simulations confirm the robustness of AGG to sparsity and sample asynchronicity, thus making it well suited for real-world applications involving incomplete multi-channel time-series data.

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

## A APPENDIX: ASYNCHRONOUS GRAPH GENERATORS

### A.1 EXPERIMENTAL DATASETS

In this section the reader can find more information regarding the three widely used benchmarks for multivariate time-series datasets that where used to compare the Asynchronous Graph Generator to other state-of-the-art works in Section 5.

#### A.1.1 BEIJING AIR QUALITY DATASET

The air quality dataset, consists of PM2.5 measurements from 36 monitoring stations in Beijing. The measurements are hourly collected from 2014/05/01 to 2015/04/30. Overall, there are 13.3% values are missing. For this dataset, we do pure imputation task with varying data removal and for the prediction task we do 6 hour prediction. Unlike the other datasets there is an explicit train/test split used in all prior works, which we followed as well in order to maintain comparable results (Cao et al., 2018; Chen, 2017; Luo et al., 2018; 2019; Silva et al., 2012; Yi et al., 2016), i.e., we use the 3rd , 6th , 9th and 12th months as the test data and the other months as the training data as outlined in Yi et al. (2016).

#### A.1.2 PHYSIONET 2012 ICU DATASET

The ICU mortality prediction health-care data used in the PhysioNet Challenge 2012 (Silva et al., 2012), consists of 4000 multivariate clinical time series from intensive care unit (ICU). Each time series contains 35 measurements such as Albumin, heart-rate etc., which are irregularly sampled at the first 48 hours after the patient's admission to ICU. We stress that this dataset is extremely sparse. There are up to 78% missing values in total. We performed the imputation task with varying additional data removed as well as the post-imputation classification task.

#### A.1.3 UCI LOCALIZATION FOR HUMAN ACTIVITY DATASET

The UCI localization data for human activity (Kaluža et al., 2014) contains records of five people performing different activities such as walking, falling, sitting down etc (there are 11 activities). Each person wore four sensors on her/his left/right ankle, chest, and belt. Each sensor recorded a 3-dimensional coordinates for about 20 to 40 millisecond. The dataset was used for the imputation task as well as post-imputation activity classification.

### A.2 ENCODER INPUT BLOCK CONSTRUCTION

In this section the reader can find more detailed instructions on how the data was deconstructed for data imputation as well as more detailed information regarding how the stride length affects the sample construction, which is directly related to data augmentation which is discussed in Section 6.

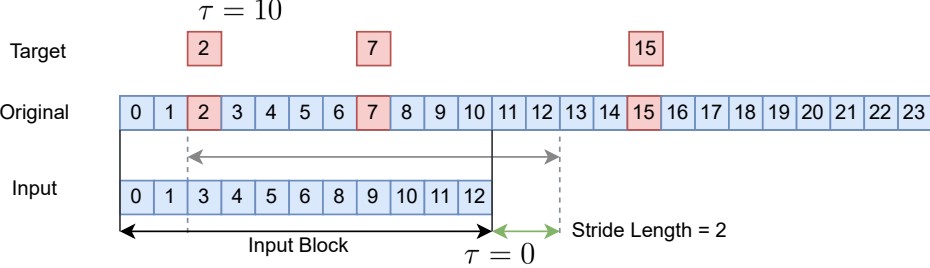

Figure 5: A pictographic representation of how the time series data is converted into an input block and a imputation target after data is randomly removed. The stride, as depicted in the image is defined as the number of steps the block is moved before it is considered the next input to the AGG. In this image the stride has value of 2 and input block a size of 11.

When building the imputation training dataset for the AGG we use the following procedure:

1. Randomly remove $r\%$ of the data, where $r \in \{10, 30, 50, 70, 90\}$

2. For the inductive imputation (KDD) all removed data that falls in the validation range defined by Yi et al. (2016) is considered validation targets, the rest is considered training, similarly for the input data. For transductive imputation (Physionet and UCI datasets) randomly select from the removed data 20% for the validation targets, the rest is considered the training targets.

3. The remaining data that was not removed is considered as the inputs and is sorted temporally.

4. As depicted in Figure 5, given a predefined context length (input block length), if there is a target (e.g. value at $t = 2$) within the range of the input block, $0 \leq t \leq 12$ then it is considered a valid target for imputation, relative time value $\tau$ is based on the largest value in the input block ($t = 12 \implies \tau = 12 - t = 0$). The input along with that single target is considered one sample with $\tau = 12 - 2 = 10$. Samples are generated for the same input block for all targets in the input range e.g. $t = 7$, where each target and input constitute an independent sample.

5. Once all targets have been coupled with inputs the block is shifted by the stride length and the process is repeated until the end of the original input is reached.

## A.3 SENSITIVITY

In this section the reader can find more information regarding why in the stride value elaborated in Appendix A.2 will directly affect the proportion of samples generated by the augmentation as seen in Figure 6a.

We can see in Figure 6b, that the smaller the stride value the better the performance because, as we can see in Figure 6a, the smaller the step the greater the training size. This large set of training examples will improve the performance up to limit, which is discussed in Section 6 in more detail.

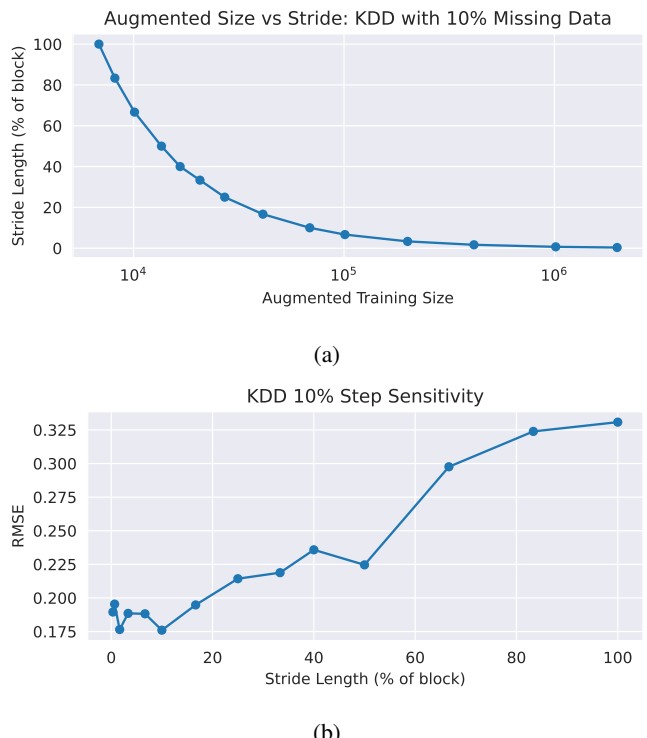

Figure 6: (a) Augmented training size vs stride as a percentage of block size. (b) Stride sensitivity.

Table 3: Validation performance for PM2.5 (Beijing dataset)

| Prediction horizon (hours ahead) | PM2.5 RMSE (over validation set) |
|---|---|
| 1 | 0.387 |
| 2 | 0.409 |
| 3 | 0.455 |
| 4 | 0.453 |

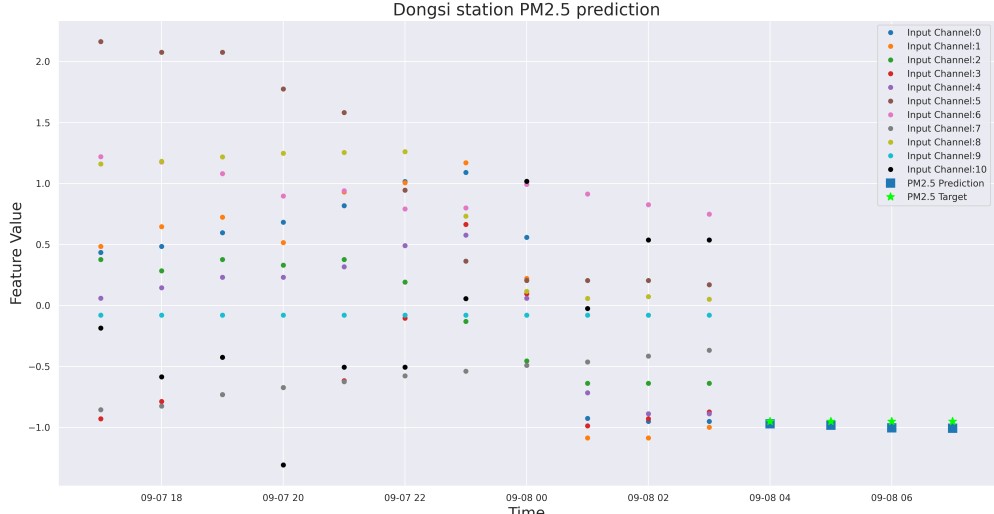

Figure 7: PM2.5 $\{1, 2, 3, 4\}$-hour ahead PM2.5 prediction. Coloured dots indicate channel inputs for the given context window, squares indicate predicted values by AGG for various hours ahead, and stars indicate the ground-truth PM2.5 values.

## A.4 PREDICTION

We evaluated the prediction performance of the pre-trained model (30% data removed imputation task) on the Beijing validation set. Table 3 shows the RMSE for the $n$-hour ahead prediction task with $n \in \{1, 2, 3, 4\}$ for the entire validation set. All setting remains the same as in the imputation task in Sec. 5. Figure 7 shows the PM2.5 prediction for the particular case of the *Dongsi* station, where dots indicate input channels, green stars are the hidden values of PM2.5 and the blue squares denote the predicted PM2.5 values. We emphasise that, as shown here, prediction is straightforwardly achieved by conditioning the generation on a *future* time value that succeeds that of the input; furthermore, this required no modification or update of the trained AGG described in the main body of the paper.

In the same line as the general imputation results shown in Sec. 5, AGG as outperformed the benchmarks in this prediction setup too. Based on the results reported by (Miao et al., 2021, Fig. 4), AGG provides lower prediction error (RMSE) than SSGAN (Miao et al., 2021), BRITS (Cao et al., 2018), GP-VAE (Fortuin et al., 2020) and NAOMI (Liu et al., 2019).

