# OpenReview forum: "Asynchronous Graph Generators"
_ICLR.cc/2024/Conference — Submitted to ICLR 2024_

### Official Review · Reviewer_QMJ3 · 2023-10-23

**Soundness:** 2 fair
**Presentation:** 2 fair
**Contribution:** 2 fair
**Rating:** 3
**Confidence:** 4

**Summary:**

This paper introduces the Asynchronous Graph Generator (AGG), a new graph neural network architecture designed for multi-channel time series data. AGG represents observations as nodes within a dynamic graph, enabling it to perform data imputation through transductive node generation without relying on recurrent components or assumptions about temporal regularity. It incorporates measurements, timestamps, and metadata as learnable embeddings within the nodes and employs attention mechanisms to capture expressive relationships among these variables. AGG implicitly learns a causal graph representation of sensor measurements, which can be used to predict new measurements based on unseen timestamps and metadata. The text discusses comparisons with previous work and the positive impact of data augmentation, highlighting AGG's state-of-the-art performance in time series data imputation, classification, and prediction across benchmark datasets like Beijing Air Quality, PhysioNet Challenge 2012, and UCI localisation.

**Strengths:**

The paper's attempt to address the data imputation problem from the perspective of an asynchronous graph is somewhat novel.

**Weaknesses:**

My main concern is that the proposed AGG architecture appears to be a fusion of various components, including transformers and graph encoders, but the paper lacks a detailed justification for why these specific combinations were chosen.
The experimental results presented in the paper appear to be somewhat limited and not entirely convincing. While AGG is shown to outperform existing methods in certain scenarios, a more comprehensive evaluation, including a broader range of datasets and potentially varying levels of data complexity, would bolster the paper's credibility. Additionally, insights into the computational resources required for AGG and any potential scalability challenges should be addressed. A more detailed discussion of the trade-offs between model complexity and performance gains would also be valuable for readers seeking to understand the practical implications of implementing AGG in real-world applications. Overall, expanding the experimental analysis and providing a more nuanced discussion of the results would further strengthen the paper's contributions and impact.
In addition, there are opportunities to enhance the paper's presentation. For instance, there could be clearer explanations of the acquisition system's workings, the data preparation process, and the interpretation of colors (e.g., green and yellow) in Figure 2.

**Questions:**

NA

---

> ### Author Response · Authors · 2023-11-14
> **Response to Reviewer QMJ3 (Weaknesses part)**
>
> **The proposed AGG architecture is a fusion of various components, including transformers and graph encoders, lacking justification for their choice.** Justification of the modules chosen by AGG has by no means been arbitrary, we tried to be as clear as possible regarding their choice throughout the paper. We succinctly refer to this here:
>
> - First 4 paragraphs of Sec 3 justify the use of an asynchronous continuous-time dynamic graph
> - Sec 3.1 presents the problem formulation (the input/output variables) and the role of the components chosen
> - Sec 3.2 presents the embeddings used:
>     - The choice of learnable Fourier-based temporal embeddings is justified from the point of view of (Fourier) spectral analysis, meaning that these embeddings will capture oscillatory information in the data
> 	- The high-dimensional (common) metadata and value embedding follow the rationale of using a homogeneous graph instead of a heterogeneous one
> - Secs 3.3 and 3.4 present the self-attention and cross-attention stages. Here, we clarified that the objective is to produce a data representation shaped to address the imputation problem.
> - Additionally, Secs 6 and 7 emphasise that inspiration for the attention stages come from successful transformer models in both language and vision applications
>
> **More experiments** As stated to Reviewer cjtX, we acknowledge that the more experiments the better. However, we have considered exactly the same setup of the SOTA and replicated their experiments in addition to perform an ablation study for data augmentation in Sec 6. _See the list of SOTA works in the response to Reviewer cjtX._
>
> **Computational resources & scalability.** We have described in detail the computational resources used at the beginning of Sec 5 under the title **Infrastructure**. Furthermore, the proposed AGG, which has a quadratic cost depending on the context length (in our case $L=100$), which results from the dot product attention computing the $L \times L$ weighted graph, it only features 378k learnable parameters and thus can run in most consumer hardware
>
> **Acquisition system and data preparation.** Thanks for this question: AGG is can deal with the data _as is_ and one of its main features is its asynchronicity: there is no need of conditioning or repairing the dataset before use. Conceptual dataset preparation (for training) is presented in detail in Sec 3.1.

---

### Official Review · Reviewer_cjtX · 2023-10-31

**Soundness:** 2 fair
**Presentation:** 3 good
**Contribution:** 1 poor
**Rating:** 3
**Confidence:** 3

**Summary:**

The paper proposes an asynchronous graph generator (AGG) for data imputation, classification, and prediction that can leverage knowledge learned from the observed asynchronous time-stamped data. Specifically, AGG predicts new sensor measurements conditioned on timestamps and metadata by adding new nodes to the learned graph.

**Strengths:**

1. Overall, the main idea of AGG is well-delivered. The problem is well-defined, and the model architectures are clearly presented.
2. The problem that the paper aims to work on of modeling asynchronous events on graph is fascinating and important.

**Weaknesses:**

1. Although the problem itself is interesting, the contribution of the paper seems to be incremental. For instance, if I understand right, the model can only predict/impute measurement (i.e., $y$). However, an essential question in asynchronous event modeling is how to effectively predict the time/attribute of the next or future events.
2. The model architecture is not novel. Most of the paper discusses embedding and attention structure, which have been well-studied in previous works. I don't think the contribution
3. More numerical experiments should be expected to highlight the effectiveness/characteristics/benefits of the model, e.g., simulation or ablation studies. The current results seem to be a black box.

**Questions:**

1. Only data imputation is illustrated in Fig. 2, and based on that, it is a bit hard to imagine the situation of prediction (i.e., new nodes come after known temporal embedding). For example, if $L=4$ in Fig. 2, which batch should nodes 5 and 6 go to?
2. Is there any real example in the experiments about prediction?
3. It seems that there is no generative mechanism going on in the model? Please clarify, if possible, the meaning of "generator" in the paper.

---

> ### Author Response · Authors · 2023-11-14
> **Response to Reviewer cjtX 1/2 (Weaknesses part)**
>
> **Contribution is incremental:**
>
> We respectfully disagree with this statement. Our work presents a thorough revision of the SOTA on graph NNs for time series and position our contribution very clearly. We summarise our contributions again here:
>
>  - The interpretation of time series as a directed graph which is free from assumptions of temporal regularity that typically constrain prior architectures such as RNNs---see **BRITS**: _Wei Cao, Dong Wang, Jian Li, Hao Zhou, Lei Li, and Yitan Li. Brits: Bidirectional recurrent imputation for time series. NeurIPS 31. Curran Associates, Inc., 2018._)
>  - The use of this geometric interpretation (inspired by _Michael M Bronstein, Joan Bruna, Taco Cohen, and Petar Veličković. Geometric deep learning: Grids, groups, graphs, geodesics, and gauges._) to derive a novel class of architectures that leverages this asynchronous graph through learnt graph attention (see _Emanuele Rossi, Ben Chamberlain, Fabrizio Frasca, Davide Eynard, Federico Monti, and Michael Bronstein. Temporal graph networks for deep learning on dynamic graphs. arXiv preprint arXiv:2006.10637, 2020._)
> - The application of this graph to multichannel time series, in a computationally efficient manner, by casting the problem from a heterogenous graph to a homogeneous graph via novel channel embedding (referred to as metadata embedding in the paper in Sec. 3)
> - The extension of SOTA (see _Petar Veličković. Everything is connected: Graph neural networks. Current opinion in structural biology, 79:102538, April 2023. ISSN 0959-440X. doi: 10.1016/j.sbi.2023.102538._) equating transformers to GNNs and applying this to the asynchronous representation via a novel causal temporal masking of dot product self attention
> - A novel methodology of transductive node generation via cross attention using a learnable representation of metadata and temporal embeddings. Time and channel can be chosen arbitrarily to conditionally _generate any_ node in the asynchronous graph allowing a flexible methodology for imputation and prediction (simply choose a time that succeeds the final measurement to cater for prediction)
>
> **The model can only impute measurement but not predict time/attribute future events.** Perhaps the aim of our work was not clear to the Reviewer: As stated in the first lines of the paper, our work focuses on data imputation and not on predicting when the next measurement will occur. This setup is not unique to our work but shared by all methods considered in the paper and the SOTA against which we compare:
>
> - SSGAN: _Xiaoye Miao, Yangyang Wu, Jun Wang, Yunjun Gao, Xudong Mao, and Jianwei Yin. Generative
> semi-supervised learning for multivariate time series imputation. In AAAI, volume 35, pp. 8983–8991, 2021._
> - BRITS: _Wei Cao, Dong Wang, Jian Li, Hao Zhou, Lei Li, and Yitan Li. Brits: Bidirectional recurrent imputation for time series. In NeurIPS 31. Curran Associates, Inc., 2018._
>
>
> **Architecture is not novel, embedding and attention have been well-studied in previous works.** We would to respectfully invite the Reviewer to be more open-minded about what a contribution to ML research is. Currently, most neural architectures are based on the attention mechanism, which has enabled clear advantages in many applications. Saying that using attention is not novel is equivalent to saying that any architecture using convolutional layers, skip connections or recurrences are invalid since those components are known. Furthermore, our work is not aimed at *studying attention and embeddings*, but to design and extend specific instantiations of these concepts that are useful to our setup. For instance: i) the temporal (Fourier-based) embedding aims to extract spectral information, ii) the metadata embedding allows flexible (homogeneous) representations of heterogeneous graphs (those with multiple sources), and iii) the attention heads aim to learn the graph dependencies among the nodes values, timestamps and metadata  (see Sec 3).
>
> **More experiments expected such as simulation or ablation studies. The current results seem to be a black box.** We acknowledge that the more experiments the better, but we do not understand what model features need to be clarified via more experiments in the view of the Reviewer, as they were not specified. We have considered exactly the same setup of all the SOTA benchmarks considered (listed above) and replicated their experiments, while also performing an ablation study for data augmentation in Sec 6. Lastly, we are unsure how to take the comment of our model being a _black box_: we have detailed as much as possible the architecture and training procedure. Perhaps our model is a _black box_ as much as any other neural network model.

---

> ### Author Response · Authors · 2023-11-14
> **Response to Reviewer cjtX 2/2 (Questions part)**
>
> **How to perform prediction.** The AGG performs prediction/imputation as transductive node generation; this is achieved by conditioning the cross-attention block on an arbitrary selection (user-defined) of the metadata and time. The selections chosen by the user are encoded via the learned embedding representation $(t2v(\tau), embed(m_t))$ (eq 8) and then used in the cross attention (Sec 3.4) AGG block along with the embeddings of the input graph to compute (predict) the value of the new node in the asynchronous graph. Furthermore, as stated in 3rd paragraph of Sec 3 in the paper: _"...If the encoding places the new node within the temporal “neighbourhood” of the other nodes in the graph, we refer to data imputation, whereas if the new node comes after the known temporal encodings we refer to prediction"_. Explicitly, if a time value is chosen that _succeeds_ the inputs, we perform prediction. Following the question of the Reviewer, we have included a prediction example on the Beijing dataset in Appendix A.4, emphasising that this required no modification of the existing architecture.
>
> Regarding the Reviewers question about context length in Figure 2, extending the context length to $L=4$ would use nodes $0,1,3,4$ in the input. Also, generating node 2 would be considered _imputation_, while generating nodes 5 and 6 would be considered _prediction_. Alternatively, nodes 7,8,10,11 could be used to generate both nodes 5 and 6, in which case we would refer to _imputation_. Note that the flexible representation of the conditional generation allows this type of arbitrary generation of nodes (which represent new samples). Note the context length of $L=3$ is simply demonstrative, the context length is a hyperparameter and is typically chosen based on the computational resources available ($L=100$ was used in our experiments).
>
> **Real world examples.** All our data come from the real datasets Beijing Air Quality, PhysioNet Challenge 2012, and UCI localisation. The reason we considered those datasets is because all the SOTA benchmarks considered are tested on them and thus we can directly report objective quantitative comparisons of AGG. See:
>
>
> - SSGAN: _Xiaoye Miao, Yangyang Wu, Jun Wang, Yunjun Gao, Xudong Mao, and Jianwei Yin. Generative
> semi-supervised learning for multivariate time series imputation. In AAAI, volume 35, pp. 8983–8991, 2021._
> - BRITS: _Wei Cao, Dong Wang, Jian Li, Hao Zhou, Lei Li, and Yitan Li. Brits: Bidirectional recurrent imputation for time series. In NeurIPS 31. Curran Associates, Inc., 2018._
>
> **Clarification of generative property.** Thank you for this question. The generative feature of the AGG refers to the way it performs transductive node generation (which is used for imputation in this specific case) via conditioning the cross attention block; we detail this in an earlier response to "how to perform prediction". More details regarding the generation can be found in Sec 3.2 (for embedding representation). See Figure 3 for a pictorial representation of how these embeddings are used to condition the generation, and Sec 3.4 for the algorithmic implementation of conditional cross attention. Also, we refer the Reviewer to Figure 1b), and c) for a conceptual pictographic demonstration of the node generation, where Figure 1b) demonstrates how the time series data is encoded into a graph which is then use to generate a new node in the graph ($h_6$, missing data). The image denotes new node in red implying it was from the "red" channel and at time 4, these would be encoded via the learned embedding representation of metadata and time2vec respectively and then used in the cross attention block (eq 8) along with the input embedding to generate the new node $h_6$ in this image.

---

### Official Review · Reviewer_qi5c · 2023-11-16

**Soundness:** 2 fair
**Presentation:** 3 good
**Contribution:** 2 fair
**Rating:** 3
**Confidence:** 4

**Summary:**

The authors present a graph-based approach for temporal data imputation using cross multi-head attention. The model assumes input of time-stamped observations with metadata and value of interest for imputation and prediction. Imputation requires target timestamps and metadata (in addition to the graph context of the input data including values) to predict the output value for the targets. They demonstrate superior performance when compared to baselines, particularly those based on RNNs and assuming discrete-time observations. The main advantage of their model in the demonstrated experiments seem due to the application of multi-head attention to construct a graph representation of the time series.

**Strengths:**

1. The authors clearly outline their modeling approach, referencing the relevant literature for embedding of temporal data, metadata, and the prediction values. The transformer encoding is intuitive and powerful, naturally inducing a continuous-time graph representation of the time series.
2. The flexibility of the approach is demonstrated methodologically in Section 4 and empirically in Section 6. The advantage of an attention-based inter-observation influence mechanism is clearly outlined as compared to RNN-type methods. The authors connect to the vision literature when investigating the data augmentation in Section 6, which is insightful.

**Weaknesses:**

1. As the key advantage of this work (when compared to the current baselines) is the use of attention, I think it is problematic that the authors have not compared extensively to attention-based temporal data imputation techniques. Particularly, [1] seems to be a very similar model and work, as they also evaluate on the same datasets. See their Table 1; AGG outperforms theirs in 10% and 50% missing data for PhysioNet, but they claim superior performance for 90%. The Beijing Air Quality data seem to be scaled differently, but you should also compare to their result.

In summary, a comparison should be made empirically and the differences in the methodology should be highlighted to other attention-based works for data imputation in temporal data.


2. The ability to impute continuous-time data is emphasized early in the paper, but then is not exploited in the experiments. While it is clear that this method outperforms discrete-time methods, I think a comparison to continuous-time imputation techniques would be insightful. For example, it seems GP-VAE does not use RNNs, and may also not require time discretization. You could hypothetically decode the Gaussian process to arbitrarily-time outputs, and compare more directly to your approach. Similarly, while there is not extensive literature on point process models for data imputation, they represent another class of continuous-time model which could be used to impute missing temporal data. For example, [2] introduces the PILES model for such a task. See also [3] Section 5.4.

In summary, I think an emphasis on the ability of your method to address continuous-time data imputation could address some limitations and similarities between other attention-based approaches. Additional experiments could highlight the true novelty of this approach, which I believe to be an attention-based method for **continuous-time** data imputation.


3. It seems a strong assumption that the time and (especially) metadata of unknown targets are known - can you demonstrate the ability to predict the metadata, at least? Regardless, I think it is a bit of a limitation that the timestamp of the imputed observation must be provided.


4. A minor comment: perhaps outlining the benefits/properties of each baseline would be helpful.



[1] Yıldız, A. Yarkın, Emirhan Koç, and Aykut Koç. "Multivariate time series imputation with transformers." IEEE Signal Processing Letters 29 (2022): 2517-2521.

[2] Chen, Jiadong, et al. "An Adaptive Data-Driven Imputation Model for Incomplete Event Series." International Conference on Advanced Data Mining and Applications. Cham: Springer Nature Switzerland, 2023.

[3] Shchur, Oleksandr, Marin Biloš, and Stephan Günnemann. "Intensity-free learning of temporal point processes." arXiv preprint arXiv:1909.12127 (2019).

**Questions:**

As above:

1. Please comment on the difference between your approach and other attention-based approaches for temporal data imputation, and compare to these apparently strong baselines.

2. Consider continuous-time models, such as a small variation of the GP-VAE baseline and perhaps a point process type of imputation model.

3. Demonstrate that you can also predict metadata given only the timestamp of a new observation.

Small questions/comments:

4. Outline the purpose and details of the baselines.

5. Line above Eq (3) “AGG is a heterogeneous graph” but it is actually homogeneous?

6. Does the MLP hidden layer need to have $l \times d_{\rm encode}$ nodes, or is this an arbitrary choice?

7. After cross multi-head attention in Figure 3, a residual connection is shown between the output of CrossMultiHead and $h_l$. However, Eq (10) shows the residual connection between the output of CrossMultiHead and $g_0$.

8. In implementation details “the input layer of dimension 5 \times 16” where does the 5 come from in this case?

---

> ### Author Response · Authors · 2023-11-21
> **AGG as a heterogeneous graph and metadata**
>
> From the reviewers comments we can see that we failed to explain how exactly the metadata is obtained and why it is valid assumption that this is known for the prediction step, this is also related to why we refer to the graph as heterogeneous and by including the metadata as an embedding we can cast the problem to a heterogeneous one. We believe a comprehensive example will help to clarify this point. The Beijing data set, for example consists of 12 weather stations seperated geographically:
> $$ station \in \\{ Aotizhongxin,Changping,Dingling, Dongsi, Guanyuan,Gucheng, \ldots, Wanshouxigong \\} $$
>
> and for each of these stations they provided different sensor measurements (from different types of sensors often simultaneously, although often their values are missing):
> $$ sensor \in \\{
>     PM2.5,
>     PM10,
>     SO2,
>     NO2,
>     CO,
>     O3,
>     TEMP,
>     PRES,
>     DEWP,
>     RAIN,
>     WSPM
> \\} $$
>  There is also additional categorical metadata for each station, this is the $WindDirection$ which is the cardinal directions
>  $$ WindDirection \in \\{NNW, E, NW, WNW, N, ENE,
>     NNE,
>     W,
>     NE,
>     SSW,
>     ESE,
>     SE,
>     S,
>     SSE,
>     SW,
>     WSW,
>     None
> \\}
> $$ The features $y$ are the measurements of any $sensor$ at any location $station$. One way to express the interconnection between physical locations and sensors is to use a heterogeneous graph, where the types of nodes are the different $station$ and $sensor$ combinations and the edges are the measurements $y$.
>
> The $WindDirection$ is a categorical feature of each node and is used to extend the information of the features through a categorical embedding. The metadata is known for all stations and sensors (as it is implicit characteristics of each node), but the measurements $y$ are not known for all sensors at all stations (as they have temporal, spatial, type dependence) and are the values we want to predict, but we need to know which $station$ and $sensor$ in order to do so, which makes sense intuitively as at time $t$ how could the model know for which $sensor$ and at which $station$ we wish to predict its value, logically the value of $PM2.5$ will vary across physical locations that are separated by many kms and many different sensor give values at the same time, thus it makes sense to _condition_ the generation step.
>
> Finally, the $timestamp$ is an independent value that is simply the time of the measurement relative to the start of the batch, there is no restriction in its value thus we can select this arbitrarily, and it is then encoded through a learnt `time2vec` embedding module (to capture frequency dynamics). Of course, we choose time values for which we have measurements so that we can train the model but generation is not bound by such restrictions.
>
> We hope this example clarifies the assumptions we make and why we refer to the graph as heterogeneous. The problem of modelling a heterogeneous graph is that we would require a different set of weights for each node type (we induces data sparsity problems), which is why we embed the node metadata and instead treat it as a homogeneous graph where the sensors type and the geographical location are encoded in the node embedding and concatenated along with the measurement $y$ (this alows the model to leverage redundancies between different node types under a common set of graphs or "attention heads"). This is also why “the input layer of dimension $5 \times 16$”, the 5 is the number of separate embedding types the feature, station, type, time and categorical (wind direction in this case) data and the 16 is the dimension of the resulting embedding. To ease the notational burden (perhaps to the detriment of clarity) we referred to station, type and categorical data collectively as "metadata".
>
> The generation step simply consists on selecting of which "metadata" we want to condition on in order to predict the $y$ value, and the model will output the generated value, there is no rational for predicting the metadata as it is simply which part of the "heterogeneous graph" we want to generate.

---

> ### Author Response · Authors · 2023-11-21
> **Weaknesses and Questions**
>
> ### Weaknesses
> 1. We thank the reviewer for connecting our work with [1], as we were not aware of this work. We will endeavor added a comparison with [1] in the revised manuscript. Your suggestion of including a more focused comparison with attention-based methods is also well taken.
> 2. We thank the reviewer for the suggestion of comparing with continuous-time imputation techniques and agree that this is a rather unexplored area of research. We will endeavor to include a comparison with GP-VAE and PILES in the revised manuscript. We also agree that the ability to impute continuous-time data is a key advantage of our method and will endeavor to highlight this in the revised manuscript, we appreciate the insight of the reviewer on this point.
> 3. Regarding the assumption that the time and metadata of unknown targets are known, perhaps we were not clear enough in the manuscript. We have provided a detailed example of how the metadata is obtained and why it is valid assumption that this is known for the prediction step in the previous section. The time of the unknown targets is simply the time of the prediction, which is arbitrary and can be selected by the user. We will endeavor to clarify this in the revised manuscript.
>
> ### Questions
> 1. Thank you for your suggestion we agree that a more comprehensive comparison is needed and will endeavor to include this in the revised manuscript.
> 2. We agree that the power of the model is in its ability to impute continuous-time data and will endeavor to highlight this in the revised manuscript.
> 3. Please refer to the previous section for a detailed example of how the metadata is obtained and why it is valid assumption that this is known for the prediction step. It makes no sense to predict the metadata as it is an implicit part of the "heterogeneous graph" we want to generate.
> 4. We will endeavor to clarify the purpose and details of the baselines in the revised manuscript.
> 5. Please refer to our response detailing the heterogeneous graph assumptions and why they are valid, we will endeavor to clarify this in the revised manuscript but we maintain that this is in fact a heterogeneous graph problem to which we cast to a homogeneous graph problem by embedding the metadata.
> 6. The MLP hidden layer does not need to have $l\times d_{\text{encode}}$ nodes, this is an arbitrary choice, we simply followed the prior work of "Attention is all you need" amoung others that expanding the hidden layer by a factor of $l$ is a good choice in order to improve the compuational capacity of the model.
> 7. Thank you for pointing this out, yes the figure is incorrect and the residual connection should be between the output of CrossMultiHead and $g_0$ as shown in Eq (10). We will endeavor to correct this in the revised manuscript.
> 8. Please refer to our detailed heterogeneous graph example for a detailed clarification. Briefly the input layer of dimension $5 \times 16$ is the number of separate embedding types the feature, station, type, time and categorical (wind direction in this case) data and the 16 is the dimension of the resulting embedding.

---

> > ### Comment · Reviewer_qi5c · 2023-11-22
> >
> > I thank the authors for the thorough response and a more clear description of the metadata. I now understand why, under this framework, prediction of metadata is not relevant.
> >
> > However, regarding a comparison to continuous-time data imputation techniques, I would say that predicting metadata may be more important. For example, one could predict all three: (1) the time of the missing observation, (2) the node on the graph at which the observation would occur, and (3) the value of the observation. I understand that this is a very different setting than the one you are considering now, but it might be an applicable area which better demonstrates the novelty of your approach, assuming you would outperform other continuous-time imputation techniques in this setting as well.
> >
> > Thank you again for the response, but I will keep my score as it is.

---

### Official Review · Reviewer_mCDq · 2023-11-16

**Soundness:** 2 fair
**Presentation:** 1 poor
**Contribution:** 2 fair
**Rating:** 3
**Confidence:** 4

**Summary:**

The work proposes a graph neural network architecture for multi-channel time-series imputation, upon leveraging embeddings and attention mechanisms. The proposed method reaches satisfactory performance on various real-data examples against baselines.

**Strengths:**

- The proposed AGG architecture is novel and useful for the imputation, classification, and regression tasks.
- The proposed method is thoroughly explained in Section 3.

**Weaknesses:**

1. Literature review: Section 2 seems to ignore more recent developments on time-series imputation in the field. See a couple mentioned in this survey paper (https://arxiv.org/abs/2307.03759) and more below. In particular, the development in the field since 2022 is barely discussed.

2. Problem setup:

- Section 3.1 introduces the problem formulation, which however is unclear to the reader. For example, is the imputation task focusing on predicting $y$? This seems to be the case as shown in Eq (12). If so, how does this differ from a standard prediction task?
- As the authors claim "no previous GNN-based method approaches the imputation problem
from the perspective of an asynchronous graph", it is important to separate alone a section explaining the formal mathematical setup of the problem, which at least contains (1) the imputation problem (2) how this is asynchronous (3) why the problem is challenging/unique that others have not proposed ways to solve it. The current Section 3.1 is highly insufficient.

3. Experiments (existing results):
- I find it strange to say "a common AGG architecture was implemented without hyper-parameter tuning for all datasets". Does this mean your method can always work without any tuning, even for learning rate/batch size, etc.? If not, it would be important to say clearly the implication behind this.
- Related to the first point, how does your method perform under various hyper-parameters, if they are actually tuned? Would it be significantly improve over current results?
- How does your model capacity compare to those of the baselines? Your model has 378k trainable parameters. How about others? What architecture of theirs is adopted in the comparison?
- The appendix should contain a table highlighting the data specifics (e.g., number of observations, number of time-series, feature dimension, etc.), as it is hard to infer these values from looking at Appendix A.1. I would suggest the authors to list these numbers in accordance with notations in previous sections. Similar thing can be done when explaining the AGG architecture.

4. Experiments (new ones currently lacking):
- The most recent baseline is SSGAN (Miao et al., 2021). However, many works have followed theirs; a quick search reveals [1-5] for the imputation task, while I believe more are existing. I thus find it unrealistic that the SSGAN is still the "state-of-the-art" method after two years.
- How does the method perform on other time-series datasets? The current experiments closely follow SSGAN, but it would be important to examine beyond that setting established more than 2 years ago.

(Incomplete) list of related papers
- [1] Miao et al., 2021: Efficient and effective data imputation with influence functions
- [2] Cini et al., 2022: Filling the G_ap_s: Multivariate Time Series Imputation by Graph Neural Networks
- [3] Alcaraz et al., 2023: Diffusion-based Time Series Imputation and Forecasting with Structured State Space Models
- [4] Liu et al., 2023: PriSTI: A Conditional Diffusion Framework for Spatiotemporal Imputation
- [5] Wu et al., 2023: Jointly Imputing Multi-View Data with Optimal Transport

**Questions:**

Questions are summarized in the weakness section above.

---

> ### Author Response · Authors · 2023-11-21
>
> We appreciate the reviewers insights into our manuscript, we will certainly take this into account for the next iteration of the article. We thank you for taking the time to produce a thoughtful response to our work and providing valuable references some of which we had not considered.

---

### Official Review · Reviewer_TnYt · 2023-11-17

**Soundness:** 2 fair
**Presentation:** 2 fair
**Contribution:** 2 fair
**Rating:** 5
**Confidence:** 2

**Summary:**

This paper studies the challenge of analyzing multi-channel time series data, particularly focusing on issues like irregular time intervals and complex spatial-temporal relationships. It proposes a novel approach with the Asynchronous Graph Generator (AGG), a graph neural network architecture that models time series observations as nodes on a dynamic graph, facilitating data imputation and prediction. AGG's unique feature is its ability to directly embed measurements, timestamps, and metadata into nodes, using attention mechanisms to discern intricate relationships among variables — which can hardly convince the reviewer — and as claimed in the paper, this method stands out from existing models by bypassing the limitations of recurrent neural networks and conventional time series models that often assume temporal regularity.

**Strengths:**

The reviewer can hardly say that they can understand the paper's method. But the idea that representing a multivariate time series as nodes in a graph is truly interesting. Even though the reviewer is not familiar with the baselines in this field, the experimental results seem relatively comprehensive and convincing.

**Weaknesses:**

The reviewer can only offer some general suggestions:

1. The reviewer believes a good research paper should be educative to general audience; they did look at Cao's work on RNN for time series imputation and find their problem set-up and  proposed approach easy to understand. Unfortunately, the current form of this paper makes it really difficult for readers without certain background knowledge to understand the setting and the contribution.

2. Given that this paper is purely empirical, the numerical experiments are the most important part to verify the performance. Table 1 may benefit from including uncertainty quantification (in the meanwhile, the reviewer acknowledges that the improvement is quite significant).

3. Terminology should be used more carefully: the term "causal" is used multiple times (and perhaps that is the reason why the reviewer gets invited to review this paper); However, it seems that "causal" merely refers to temporal order, which is "Granger causal" and means correlation from past to the future. The reviewer suggests using simple terms like temporal order directly.

4. Scaling might be one major issue when the dimension is high and the time horizon is long (since there must be a really huge graph to represent the multivariate time series).

**Questions:**

There are two major concerns that make the reviewer lean towards rejection of this work:

1. In Fig. 1 (c), the proposed method used future to predict past. The reviewer is fine with using the expressive power of neural networks to learn a latent causal representation, but a structure shown in Fig. 1 (c) seems to imply that the latent data generation mechanism depends on the future which is clearly wrong. Can the authors justify that?

(PS: in granger causal literature, one famous example is that "buying Christmas tree" Granger causes "Christmas". However, this example cannot justify the proposed structure since there should be a latent variable "knowing Christmas will be on 12/25" captured by the latent data generating process).

2. Another reason why the reviewer gets invited to reviewer this paper might be the use of Physionet dataset — the are a lot of lab tests in that dataset where missing values themselves mean that the clinician is not suspect of related dysfunction/disease at all. That is why there are so many missing values in that dataset —  a single patient cannot be suspected to have all diseases — and the missingness carries meaning. Can authors justify on why imputing this dataset?

---

> ### Author Response · Authors · 2023-11-21
>
> We appreciate the reviewer taking the time to review our work and providing a valuable general perspective, we will endeavour to include the reviewers suggestions in our next iteration of the manuscript
> ## Weaknesses
> 1.  We will endeavour to make the next iteration of this manuscript more accessible to the general audience.
> 2. Thank you for the insight we agree uncertainty quantification would improve the imputation section, for the post imputation classification and regression we performed a 5 fold cross validation which indeed quantifies the uncertainty so some degree.
> 3.  We appreciate the connection to "Granger" causality, we had not considered this definition, and the reviewer is correct the "causal" refers to the "temporal order", although the underlying design is to capture (learn through graph attention) the phase dynamics between different measurements i.e. that the measurement from sensor $y_k$ at time $t-2$ is related to sensor $y_l$ at time $t$ by some hidden dynamics (which are related by some phase shift in the signals) of which we wish to learn through graph attention.
> 4. We will endeavour to explore the limitations of architecture wrt to high dimensionality and longer time horizons, thank you for the suggestion.
> ## Questions
> 1. Figure 1b) shows the temporal ordering of the masking of the graph, which respects the ordering of the signals, this input graph is then encoded along with the signals through the AGG which results on a set of encoded signals $h$ which a used to either impute or predict any other node. In the case of figure 1c) this is demonstrating imputation (filling in missing values) which indeed uses the future value to fill in the missing sensor measurements. This is justified as the purpose is to use all available information to infer the missing value. In the prediction case this indeed would be a violation but would not occur as the generated node would be after the encoded inputs in order to be prediction. We appreciate the reviewers question.
> 2. Simply this dataset has been a benchmark for imputation for many state of the art papers in the past, including all the papers to which we compare our results, most notably SSGAN by Miao et al., so it seemed natural to apply our paper to this benchmark in order to show fair comparisons. That being said the imputation could have many benefits for this type of scenario, it may be informative to estimate for example the blood pressure of a patient at any point of time when examining the patients history in the ICU that eventually led to their death, this is obviously infeasible that these measurements were taken constantly and impossible to retroactively know the ground truth of this and other indicators. With this type of model under the imputation set up we have shown we can reasonably estimate these unknown values for various levels of sparsity

---

> > ### Comment · Reviewer_TnYt · 2023-11-22
> >
> > The reviewer has read the response and decided to keep the score.

---

### Author Response · Authors · 2023-11-14
**General response**

We thank the Reviewers for their feedback, we reply to the comments point by point. Summarised Reviewer's comments are presented in **bold font** in each particular reply.

In addition to our responses, and as suggested by the Reviewers, we have included an additional experiment in Appendix A.4, where we assess the prediction performance of AGG using the Beijing dataset. In the same line of the experiments of the main body of the paper, AGG outperformed the current SOTA in the prediction setting as per the previous results provided in the SSGAN paper.

**SSGAN paper**: _Xiaoye Miao, Yangyang Wu, Jun Wang, Yunjun Gao, Xudong Mao, and Jianwei Yin. Generative
semi-supervised learning for multivariate time series imputation. In Proceedings of the AAAI
conference on artificial intelligence, volume 35, pp. 8983–8991, 2021._

---

### Meta-Review · Area_Chair_m1Kz · 2023-12-15

**Metareview:**

This paper introduces the asynchronous graph generator (AGG), a graph neural network architecture for multi-channel time series that models observations as nodes on a dynamic graph and can thus perform data imputation by transductive node generation. Although the paper may contain some interesting ideas, it did not consider some recent literature and thus lacks comparison and possibly important related references. The paper may need further development.

**Justification For Why Not Higher Score:**

The paper miss citing and comparing with the more recent literature.

**Justification For Why Not Lower Score:**

NA

---

### Decision · Program_Chairs · 2024-01-16

Reject